# Comorbid mental disorders and quality of life of people with epilepsy attending primary health care clinics in rural Ethiopia

Ruth Tsigebrhan[1]⊗*, Abebaw Fekadu[1,2,3‡], Girmay Medhin[1,4‡], Charles R. Newton[5‡], Martin J. Prince[6‡], Charlotte Hanlon[1,2,6⊗]

1 Centre for Innovative Drug Development and Therapeutic Trials for Africa (CDT-Africa), College of Health Sciences, Addis Ababa University, Addis Ababa, Ethiopia, 2 Department of Psychiatry, WHO Collaborating Centre in Mental Health Research and Capacity-Building, School of Medicine, College of Health Sciences, Addis Ababa University, Addis Ababa, Ethiopia, 3 Department of Global Health & Infection, Brighton and Sussex Medical School, Brighton, United Kingdom, 4 Aklilu-Lemma Institute of Pathobiology, Addis Ababa University, Addis Ababa, Ethiopia, 5 Department of Psychiatry, University of Oxford, Warneford Hospital, Warneford Lane, United Kingdom, 6 Centre for Global Mental Health, Health Services and Population Research Department, Institute of Psychiatry, Psychology and Neuroscience, King's College London, London, United Kingdom

⊗ These authors contributed equally to this work.
‡ AF, GM, CRN and MJP also contributed equally to this work.
* r_tessera@yahoo.com

**Data Availability Statement:** All relevant data are within the manuscript and its Supporting Information files. The 10-item Quality of Life in Epilepsy questionnaire (QOLIE-10p) is available

## Abstract

### Background

Evidence from high-income countries demonstrates that co-morbid mental disorders in people with epilepsy adversely affect clinical and social outcomes. However, evidence from low-income countries is lacking. The objective of this study was to measure the association between co-morbid mental disorders and quality of life and functioning in people with epilepsy.

### Methods

A facility-based, community ascertained cross-sectional survey was carried out in selected districts of the Gurage Zone, Southern Ethiopia. Participants were identified in the community and referred to primary health care (PHC) clinics. Those diagnosed by PHC workers were recruited. Co-morbid mental disorders were measured using a standardised, semi-structured clinical interview administered by mental health professionals. The main outcome, quality of life, was measured using the Quality of Life in Epilepsy questionnaire (QOLIE-10p). The secondary outcome, functional disability, was assessed using the 12-item World Health Organization Disability Assessment Schedule (WHODAS-2).

### Results

The prevalence of comorbid mental disorders was 13.9%. Comorbid mental disorders were associated with poorer quality of life (Adjusted (Adj.) β -13.27; 95% CI -23.28 to-3.26) and greater disability (multiplier of WHODAS-2 score 1.62; 95% CI 1.05, 2.50) after adjusting for

from doi.org/10.1111/j.1528-1157.1996.tb00612.x., the Operational Criteria for Research (OPCRIT plus) is available from doi.org/10.1192/bjp.bp.110.082925, and World Health Organization Disability Assessment Schedule version 2.0 (WHODAS-2) is available from the WHO (https://www.who.int/classifications/icf/whodasii/en/).

**Funding:** This study was conducted as part of a Wellcome Trust fellowship for RT (Grant Number 104023/Z/14/A) and a PhD fellowship from CDT-Africa The study was nested within the PRogramme for Improving Mental health carE (PRIME). PRIME was funded by the UK Department for International Development (DfID) [201446]. The views expressed in this article do not necessarily reflect the UK Government's official policies. CH is supported by the National Institute of Health Research (NIHR) Global Health Research Unit on Health System Strengthening in Sub-Saharan Africa, King's College London (GHRU 16/136/54). The views expressed are those of the author and not necessarily those of the NHS, the NIHR or the Department of Health and Social Care. CH additionally receives support from AMARI as part of the DELTAS Africa Initiative [DEL-15-01].

**Competing interests:** The authors have declared that no competing interst exist.

hypothesised confounding factors. Low or very low relative wealth (Adj. β = -12.57, 95% CI -19.94 to5.20), higher seizure frequency (Adj.β coef. = -1.92, 95% CI -2.83 to -1.02), and poor to intermediate social support (Adj. β coef. = -9.66, 95% CI -16.51 to -2.81) were associated independently with decreased quality of life. Higher seizure frequency (multiplier of WHODAS-2 score 1.11; 95% CI 1.04, 1.19) was associated independently with functional disability.

## Conclusion

Co-morbid mental disorders were associated with poorer quality of life and impairment, independent of level of seizure control. Integrated and comprehensive psychosocial care is required for better health and social outcomes of people with epilepsy.

## Introduction

Globally, the lifetime prevalence of epilepsy is estimated to be 0.76%, with a higher prevalence seen in low and middle income countries (LMICs) (0.88%) [1]. The age-standardized disability adjusted life years (DALYs) between the year 1990–2016 were 182.6 per 100,000 population in women and 201.2 per 100,000 population in men [2]. Epilepsy can negatively affect a person's social, occupational and interpersonal functioning [3]. Persons with epilepsy may face stigma and discrimination from their communities because of the unpredictable nature of the disorder and misperceptions about the cause of the illness [3–5]. Epilepsy and the associated stigma can be a barrier to forming friendships and relationships and to employment opportunities [3].

In addition, the high prevalence of mental disorders among people with epilepsy exacerbates the impact of the disease [6, 7]. Depression and anxiety disorders are the most common co-morbid mental health conditions, with pooled prevalence of 17.8% and 8.1%, respectively, based on studies using clinical evaluation [8]. People with epilepsy are also at increased risk of a range of other mental health conditions including psychosis, bipolar disorder and personality disorders [9]. In previous research conducted in high-income countries (HIC), co-morbid mental health conditions were associated strongly with poor quality of life, increased disability, an increased suicide rate, poorer adherence to treatment and higher levels of stigma [3, 6, 7, 10–12]. In recognition of the high burden of co-morbid mental disorders, guidelines for care of people with epilepsy in HICs emphasise the importance of integrated evaluation and management of mental disorders in order to achieve better seizure control and improved quality of life [7].

Despite this evidence from HIC, very few studies from LMICs, especially sub-Saharan Africa, have been conducted on the impact of comorbid mental health conditions on disability and quality of life [13–17]. Many of the existing studies on quality of life have important limitations, such as participants being recruited from tertiary health services who may not be representative of the general population of people with epilepsy [14, 16–18], small sample sizes [13, 18] and reliance on self-report screening tools for mental disorder rather than clinical assessments [16–18].

The main objective of this study was to examine the association between co-morbid mental health conditions and quality of life and functioning in people with epilepsy in a rural area of Ethiopia. We hypothesized that comorbid mental health conditions would be independently

associated with decreased quality of life and increased disability compared to those without comorbid mental health conditions. Quantifying the contribution of mental health conditions to disability and poorer quality of life in LMICs is important to develop locally relevant interventions and protocols for providing holistic and integrated care.

## Materials and methods

### Design

A facility based cross-sectional survey of community-ascertained people with epilepsy.

### Setting

The study was conducted in four selected districts of the Gurage zone, in the Southern Nations, Nationalities, and Peoples' region (SNNPR) of Ethiopia. At the time of the last census (2007), the Gurage Zone had a total population of 1,279,646 people, of whom 622,078 were men and 657,568 women. Most (80.5%) of the Gurage population speak Gurage languages as their first language, although the majority can also converse in Amharic, which is the official language of the region [19]. Around half of inhabitants are Muslim (51.0%) [19]. The Zone is subdivided into 15 districts (woredas) [19]. For this study, four Gurage zone districts (Sodo, Eja, Kebena and Wolikette) were selected purposively, since these districts with their respective health centres have all shown tremendous efforts to integrate mental health services. Inclusion of these districts was also logistically feasible.

This study was nested within the Programme for Improving Mental health carE (PRIME), a research programme consortium across five low- and middle-income countries (LMICs); Ethiopia, Nepal, India, South Africa and Uganda [20]. From 2014 onwards, PRIME Ethiopia worked with the district health office and local stakeholders to develop and implement a programme of care for people with epilepsy, psychosis, depression and alcohol use disorders, integrated within primary healthcare services in Sodo district [21]. The district level intervention was based on the World Health Organization's mental health Gap Action Programme (mhGAP) [22], whereby health officers and nurses based in primary health care facilities were trained to assess and treat people with the selected priority conditions. The treatments included basic psychoeducation and psychosocial support, with psychotropic medication prescribed by the primary healthcare worker if indicated. PRIME undertook detailed evaluation of the programme in Sodo district [23] and, from 2016 onwards, worked with the zonal health office to scale-up the model service to other districts in the Gurage zone. The current study was conducted during the scale up phase of the PRIME project. In Sodo district (the initial PRIME implementation site), eight health centres were the sites for data collection. For the other three Gurage districts, one health centre located in each of the selected districts was the focus for collection of data.

### Study population and sampling

Case detection was carried out by key informants and health extension workers (HEWs) who had been trained to recognize people with possible active convulsive epilepsy, augmented by house-to-house screening by HEWs [21]. Screen positive individuals were referred to the nearby health centre and the diagnosis of epilepsy was confirmed by the mhGAP-trained PHC workers. This two stage screening method has been used previously [24] and was implemented in the PRIME study [23]. After confirmation of the diagnosis by the PHC worker, people with diagnosis of epilepsy were provided with treatment and follow-up care, regardless of whether they participated in this study.

Participants were eligible for the study if they provided informed consent and met the following criteria: (i) PHC worker diagnosis of active convulsive epilepsy defined based on the mhGAP definition of convulsive epilepsy: recurrent (at least twice) unprovoked seizure occurring days or weeks apart, (ii) aged ≥ 18 years, and iii) no plans to migrate out of the study district in the following 12 months. Exclusion criteria: (i) communication difficulties due to cognitive or intellectual disability, (ii) unable to converse in Amharic, the official language of Ethiopia, (iii) lacking the capacity to consent to participation in the study. A trained project psychiatric nurse assessed capacity to consent using an approach used previously in this setting [25].

**Sample size.** Sample size was calculated to detect a five point mean difference in quality of life score between people with epilepsy with and without a co-morbid mental health condition, with a significance level of 0.05, accounting for health centre (n = 4) clustering by assuming intra-cluster correlation of 0.01 [26] and allowing 20% for non-response. Assuming a prevalence of mental disorders in people with epilepsy of 35–40% [27], around 320 participants were needed to be screened for mental disorder. Ethical approval was obtained from the Institutional Review Board of the College of Health Sciences, Addis Ababa University and the Research Ethics Committee of King's College London (HR-15/16-2434). Informed written or witnessed verbal (for non-literate participants) consent was obtained. Treatment was provided for epilepsy and/or other mental health conditions at the health centre.

## Measurements and data collection

**Primary outcome.** *Quality of life*. Was measured using the 10-item Quality of Life in Epilepsy questionnaire (QOLIE-10p) [28]. This questionnaire was derived from the original longer QOLIE-89 scale [29]. The QOLIE-10p covers both general and epilepsy-specific areas, grouped into three factors: epilepsy effects (memory, physical effects, mental effects of medication), mental health (energy, depression, overall quality of life) and role functioning (seizure worry, work, driving, social limits). The questionnaire was translated into Amharic by the first author (RT), back-translated by a non-mental health professional and then consensus was reached on the final version of the questionnaire through a group discussion of experts in mental health in Ethiopia. The total mean score of the QOLIE-10p ranges from 0–100, with a higher score indicative of better quality of life.

**Secondary outcomes.** *Functional disability*. Was measured using the 12-item version of the interviewer administered World Health Organization Disability Assessment Schedule version 2.0 (WHODAS-2) [30]. The WHODAS-2 has been validated in people with chronic diseases, including epilepsy [31], and the Amharic version has been shown to have convergent and predictive validity in a rural Ethiopian setting [32, 33]. Higher scores on WHODAS-2 indicate a higher degree of functional impairment.

**Primary exposure.** *Clinician-diagnosed mental health condition*. Comorbid mental disorder was defined as the presence of any coexisting psychiatric disorders [9] based on the research criteria for the Diagnostic and Statistical Manual for Mental Disorders (DSM-IV) or the International Classification of Disorders (ICD-10) in addition to diagnosis of convulsive epilepsy. The Operational Criteria for Research (OPCRIT plus paper version) is a semi-structured clinical interview and a psychopathological checklist. It was completed by trained psychiatry nurses in the original language (English) [34]. Software are used to run diagnostic algorithms on the completed assessments. The OPCRIT+ has been shown to have good interrater reliability in a study from United kingdom (weighted kappa of 0.70 for diagnostic reliability) [34].

**Potential confounding variables.**

- Socio-demographic characteristics: age, sex, education, marital status, area of residence and relative wealth measured by subjective account of wealth compared to the other people in the society of the participant.

- Seizure frequency: The number of seizures in the month prior to data collection.

- Duration of epilepsy

- Social support was measured using the Oslo Social Support scale (OSS-3) [35], which is a three item questionnaire to assess the perceived social support of the participants. The total score ranges from 3–14. A higher score indicates better social support. The OSS-3 has been used in several studies in Ethiopia including in Gurage zone [36], with good predictive and convergent validity [37].

**Other psychosocial risk factors predicted to be on the causal pathway.**

- Stressful life events: experience of stressful life events was assessed using the List of Threatening experiences (LTE) adapted for the context [38]. The LTE has 12 categories of stressful events which are rated for the six months prior to data collection.

- Perceived stigma was measured using the stigma section of the Family Interview Schedule (FIS) questionnaire [39]. This instrument has been translated into Amharic and used previously in rural Ethiopia to measure stigma in people with epilepsy and those with mental health conditions [4, 40].

## Data collection

The confirmatory clinical evaluations of epilepsy were completed by primary healthcare professionals. The OPCRIT assessments were carried out by psychiatric nurses who had been previously trained and had experience on the administration of the instrument. All other measures were administered by lay interviewers educated to a minimum of tenth grade (completion of secondary school). The lay data collectors were trained on the administration of the instruments for one week by the principal investigator and then carried out several practice interviews. Close supervision was provided by the project employees and psychiatric nurses.

## Data analysis

Data were double entered using Epi-data version 3.1 and analysed using STATA version 12. Simple descriptive analyses (mean, median and percentages) were used to summarise the socio-demographic and clinical characteristics of the study participants.

Confirmatory factor analysis was conducted using the AMOS software version 23 for the quality of life measure (Amharic version of QOLIE -10p) against the original three factor structure of QOLIE-10p [28]. The overall goodness of fit of the model was measured by the Root Mean Square Error Approximation (RMSEA), Tucker-Lewis Index (TLI) and Comparative Fit Index (CFI). The proposed three structure did not fit the data well and exploratory factor analysis (EFA) with maximum likelihood extraction and varimax rotation was done. The scree plot and eigenvalues were used to determine the number of sub-scales.

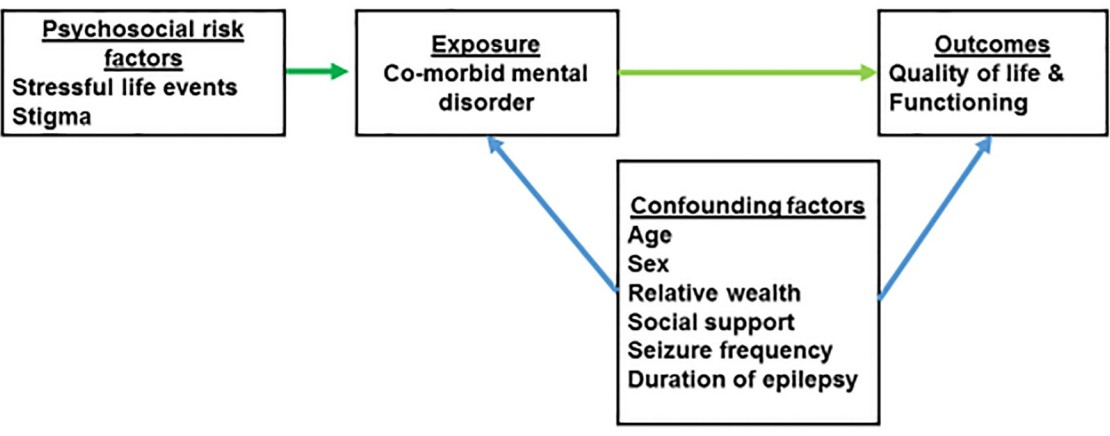

**Fig 1. Conceptual model.**

A hypothesis-driven analysis was carried out based on the conceptual model (see Fig 1). Univariate analysis followed by multiple linear regression modelling was used to examine the association between quality of life and the primary exposures, adjusting for all potential confounding factors identified a priori.

A sensitivity analysis was carried out, in which the mental health items of the QOLIE-10pmeasure were excluded.

The polytomous scoring method was used to calculate the total score of WHODAS. As the score of WHODAS was positively skewed and over-dispersed, negative binomial regression was used to examine the association between clinical and sociodemographic factors and functional disability. Multiple regression was also repeated after including the factors that were hypothesised to be on the causal pathway (stressful life events and epilepsy-related stigma) between co-morbid mental disorder and quality of life/functional disability) to explore possible mediation.

## Results

### Sociodemographic and clinical characteristics

A total of 246 people with epilepsy attended the health centres of the selected four districts for evaluation and 239 (96.3%) of them were recruited over a period of 11 months (March 2017-January 2018). Full data were available on 237 participants. Nine participants were excluded from the study: the data of two participants were incomplete and seven did not fulfil the eligibility criteria. Most of the participants (84.4%) were from the district of Sodo.

The median age of the participants was 30 (inter-quartile range (IQR) = 22–42 years) and there were more males (59.1%) than females. Most resided in a rural area, reported a low or very low income (subjective assessment of wealth) and had not received formal education. Half of the participants were married (51.9%) and were farmers (47.3%). The median family size was 5 (IQR = 4–7). Nearly two-thirds rated their social support as poor to intermediate. Exposure to psychosocial stressors was high: nearly half of participants had experienced a stressful life event in the past six months and 82.2% of them had experienced one or more of the items indicating perceived stigma (Table 1).

All participants were diagnosed with generalised seizure types. Only 10.5% of participants had not received any biomedical treatment prior to recruitment into the cohort. The median number of seizures in the month prior to data collection was one.

**Table 1. Sociodemographic characteristics.**

| Characteristics (Total N = 237) | | Number (%) |
|---|---|---|
| Median age (IQR) | | 30 (22–42) |
| Sex | Male | 140 (59.1) |
| | Female | 97 (40.9) |
| Marital status | Never married or formerly married | 114 (48.1) |
| | Married | 123 (51.9) |
| Education | No formal education | 135 (57.0) |
| | Formal education | 102 (43.0) |
| Employment | Employed | 21 (8.9) |
| | Unemployed | 15 (6.3) |
| | Farmer | 112 (47.3) |
| | House wife | 61 (25.7) |
| | Others * | 28 (11.8) |
| Relative wealth | Low or very low | 169 (71.3) |
| | Average and above | 68 (28.7) |
| Area of residence | Rural | 208 (87.8) |
| | Urban | 29 (12.2) |
| Religion | Orthodox Christian | 208 (87.8) |
| | Protestant | 15 (6.3) |
| | Muslim | 7 (2.4) |
| Median family size (IQR) | | 5 (4–7) |
| Number of stressful life events in the past 6 months | None | 129 (56.8) |
| | 1 or 2 | 69 (30.4) |
| | 3 and above | 29 (12.8) |
| Social support | Poor- intermediate | 144 (63.4) |
| | High | 83 (36.6) |

* includes students and other jobs

IQR: Interquartile Range; SD: standard deviation

Based on the psychiatric nurse clinical diagnosis, 13.9% (n = 33) of the participants had a comorbid mental disorder, of which major depressive disorder (MDD) was the most common comorbid disorder. The mean quality of life score was 69.7 (SD = 19.3) and the median number of days spent with difficulty accomplishing usual activities and work was 5 (IQR 2–8) days of the past 30 days (Table 2).

## Quality of life

The fit indices of the three factor structure of QOLIE-10p resulted were as follows: $\chi^2$ = 195.89, (d.f = 32;p<0.0001), CFI = 0.85, TLI = 0.79 and RMSEA = 0.15, indicating inadequate fit of the model to the data. EFA showed a unidimensional factor structure with two items (seizure worry and trouble with driving /transportation) loading low (< 0.35). Since dropping of two items makes the instrument non valid, the total weighted mean score of all the items was used for analysis.

In the hypothesis-driven analysis, comorbid mental disorders were associated with decreased quality of life in both univariable (β = -17.04, 95% confidence interval (CI) -26.9 to -7.2) and multivariable analysis (Adj. β = -13.27, 95% CI -23.28 to-3.26). In the sensitivity analysis that excluded the mental health items of the QOLIE-10p, the result was similar.

**Table 2. Clinical characteristics of study participants.**

| Clinical characteristics | Number (%) |
|---|---|
| Previous biomedical treatment (lifetime) | 212 (89.5) |
| Median age of epilepsy onset (IQR) | 17 (10–27) |
| Median duration of epilepsy in years (IQR) | 11 (0–40) |
| Median seizure frequency (IQR)/month | 1 (0–2) |
| Mean Quality of Life score (SD) | 69.7 (19.3) |
| Median number of days with disability in the past month (IQR) | 5 (2–8) |
| Median WHODAS-2 score (IQR) | 11.1 (5.6–27.8) |
| Diagnosis of comorbid mental disorder | |
| None | 204 (86.0) |
| Major Depressive Disorder | 18 (7.6)* |
| Dysthymia | 2 (0.8) |
| Psychosis | 5 (2.1) |
| Alcohol Use Disorder | 7 (3.1) |
| Bipolar disorder | 2 (0.8) |

SD- Standard deviation, IQR- Interquartile range,

*one participant had Major Depressive Disorder + Alcohol Use Disorder.

Low or very low relative wealth (Adj.β coef. = -12.57, 95% CI -19.94 to -5.20), higher seizure frequency (Adj.β coef. = -1.92, 95% CI -2.83 to -1.02), and poor to intermediate social support (Adj. β coef. = -9.66, 95% CI -16.51 to -2.81) were also associated with decreased quality of life after adjusting for the hypothesized confounding factors. See Table 3.

When comorbid mental disorders and seizure control was entered to the model separately, variation in quality of life was more explained by seizure control (adj. $R^2$ = 16%) than comorbid mental disorders (Adj. $R^2$ = 9.5%).

Epilepsy related stigma (Adj.β coef. = +4.77, 95% CI +1.97 to +7.56) was associated with comorbid mental disorders but stressful life events (Adj.β coef. = +0.43, 95%CI -0.18 to +1.04) was not associated with comorbid mental disorders. After entering epilepsy-related perceived

**Table 3. Sociodemographic and clinical characteristics associated with quality of life (weighted QOLIE-10p score).**

| Characteristic | | Univariate analysis | | Multivariable analysis | |
|---|---|---|---|---|---|
| | | Crude β coef. | 95% CI | Adjusted β coef. | 95% CI |
| **Age (years)** | | -0.24 | -0.52 to +0.03 | -0.23 | -0.54 to-0.66 |
| **Gender** | Male | 1 | 1 | 1 | 1 |
| | Female | -1.14 | -8.32 to +6.03 | -0.61 | -7.50 to +6.28 |
| **Relative wealth** | Average and above | 1 | 1 | 1 | 1 |
| | Very low &low | -11.83 | -19.49 to-4.18 | -12.57 | -19.94 to—5.20 |
| **Education** | No formal | 1 | | 1+0.65 | 1 |
| | Formal education | 2.65 | -4.47 to +9.77 | | -6.48 to +7.78 |
| **Seizure frequency/ month** | | -2.00 | -2.92 to -1.07 | -1.92 | -2.83 to-1.02 |
| **Duration of epilepsy/ years** | | -0.28 | -0.63 to +0.06 | -0.25 | -0.61 to 0.11 |
| **Comorbid mental disorder** | No | 1 | | | 1 |
| | Yes | -17.04 | -26.92 to-7.15 | -13.27 | -23.28 to -3.26 |
| **Social support** | Strong | 1 | 1 | 1 | 1 |
| | Poor- intermediate | -7.99 | -15.36 to—0.61 | -9.66 | -16.52 to -2.81 |

**Table 4. Sociodemographic and clinical characteristics associated with functional disability (total WHODAS score).**

| Characteristic | | Univariate analysis | | Multivariable analysis | |
|---|---|---|---|---|---|
| | | Multiplier of WHODAS-2 score | 95% CI | Multiplier of WHODAS-2 score | 95% CI |
| **Age** | | 1.00 | 0.99, 1.01 | 1.01 | 1.00, 1.02 |
| **Gender** | Male | 1 | 1 | 1 | 1 |
| | Female | 1.23 | 0.91, 1.65 | 1.30 | 0.97, 1.75 |
| **Relative wealth** | Average and above | 1 | 1 | 1 | 1 |
| | Very low &low | 1.13 | 0.82, 1.56 | 1.25 | 0.90, 1.72 |
| **Education** | No formal | 1 | 1 | 1 | 1 |
| | Formal education | 1.10 | 0.82, 1.47 | 1.11 | 0.83, 1.50 |
| **Seizure frequency/ month** | | 1.13 | 1.06, 1.21 | 1.11 | 1.04, 1.19 |
| **Duration of epilepsy/years** | | 1.00 | 0.99, 1.02 | 1.00 | 0.99, 1.02 |
| **Comorbid mental disorder** | No | **1** | 1 | 1 | 1 |
| | Yes | 1.83 | 1.21, 2.76 | 1.62 | 1.05, 2.51 |
| **Social support** | Strong | 1 | 1 | 1 | 1 |
| | Poor- intermediate | 1.04 | 0.76, 1.42 | 1.10 | 0.82, 1.48 |

stigma and stressful life events into the multivariate model, comorbid mental disorders were no longer associated significantly with quality of life (Adj. β coef. = -5.26, 95% CI -14.11 to +3.58).

## Functional disability

Results obtained from the univariate and multivariable models that modelled functional disability are presented in Table 4. Comorbid mental disorders were significantly associated with increased functional disability (multiplier of WHODAS-2 score 1.83; 95% Confidence Interval (CI) 1.21, 2.76). After adjusting for hypothesized confounders, having comorbid mental disorders (multiplier of WHODAS-2 score 1.62; 95% CI 1.05, 2.50) and higher seizure frequency (multiplier of WHODAS-2 score 1.11; 95% CI 1.04, 1.19) were independently and significantly associated with functional disability. When epilepsy-related stigma and stressful life events were included in the multivariable model, comorbid mental disorders were no longer significantly associated with functional disability (multiplier of WHODAS-2 score of 1.28, 95% CI 0.88, 1.86).

## Discussion

In this study, we investigated comorbid mental disorders in people with epilepsy and their association with quality of life and functional disability. We found evidence to support our hypothesis that comorbid mental health conditions with epilepsy were associated with both poorer quality of life and impaired functioning. Seizure frequency was also associated independently with poor quality of life and impaired functioning. Moreover, lower quality of life was significantly associated with very low relative income and intermediate to poor social support.

The prevalence of comorbid mental disorders in people with epilepsy was high in this rural Ethiopian setting compared to the general population [41] but lower than studies conducted in high income countries [8]. The lower prevalence of comorbid mental disorders could be due to the study setting. This study was done in a primary care setting whereas people with more severe forms of epilepsy (and higher risk of co-morbid mental health problems) could have been referred to secondary or tertiary health care. The low prevalence could also be due to the generally low levels of detection of common mental disorders in rural Ethiopia [42]. The

way in which depression and anxiety manifest and the non-biomedical causal attributions of depressive/anxiety symptoms in this socio-cultural context [43] could have contributed to the low detection by mental health professionals applying Western diagnostic criteria [42]. The diagnosis and presentation of depression in Ethiopian culture appears to include a combination of anxiety, somatic and depressive symptoms rather than typical DSM criteria of depression [44]. This might have masked and contributed to the absence of diagnosis of anxiety disorders in this study. Psychiatry nurses could also be biased towards diagnosing more severe conditions like depression than anxiety disorders in their clinical practice.

The association between comorbid mental disorders and poor quality of life is consistent with the findings of previous studies carried out in both high-income and LMICs [7, 12, 13, 17]. Higher seizure frequency has also been shown to be associated with poor quality of life [12], although some studies found that comorbid mental disorders existed despite good control of seizures [12]. Quality of life can be affected by a range of clinical and psychosocial factors. Jacoby et al. [11] have categorized these factors as epilepsy-related and not epilepsy-related. From the non-epilepsy-related factors, comorbid mental and physical disorders, stigma, resilience, self-efficacy and social support have all been identified as important [11]. Even though both seizure control and comorbid mental disorders play a significant role in the quality of life, the findings of this study support the negative contribution of mental disorders upon quality of life irrespective of seizure control.

The presence of epilepsy-related stigma may have mediated the relationship between the exposure (comorbid mental disorder) and the outcome (quality of life) in this study. Stigma has a significant negative consequence on the lives of people with epilepsy, for example in terms of social life and work, which, in turn, can affect psychological wellbeing [11]. Stigma may also lead to mental ill-health through the acceptance and internalization of the societal devaluation of the individual [5]. Stigma may lead to a low sense of self-efficacy and decreased access to social support, which in turn may lead to less optimal self-management of epilepsy, higher seizure frequency and poorer quality of life [11]. Although a causal pathway linking stigma to poor quality of life could not be determined given our cross-sectional study design, the strong association of stigma with quality of life after adjusting for comorbid mental disorders is in keeping with our *a priori* conceptual model.

Poorer relative wealth was associated with quality of life. This association was not found in some studies done in Africa [17, 45]. Both these studies assessed wealth by monthly monetary income but the study from Ethiopia [17] has found an association of lower education level with poor quality of life. The measurement of quality of life in a poverty setting is a complex construct but the definition of quality of life includes 'an individual perceptions of their position in the context of the culture and value system in which they live, and in relation to their goals, expectations, standards and concerns' [46]. The high levels of stigma in this study population may be a major barrier to achieving individual goals and achieving the standards of the community. Perceived stigma may affect opportunities for education and employment, which in turn results in poor income. The low perception of self-position in the society in addition to the poverty then impacts on quality of life. Furthermore, measurement of wealth status used in this study was done in comparison to the other member of the society which also may suggest poor self-esteem which is associated with poor quality of life.

Functional disability was found to be associated independently with comorbid mental disorders and seizure frequency. These findings are similar to a study carried out in Canada [10]. Co-morbid depression has been shown to be associated with the greatest decrement in the health status of an individual for a range of different chronic disorders, for example asthma and diabetes, compared to the chronic physical health condition alone [47]. Despite the high prevalence of epilepsy in low income countries compared to HICs and its associated neuro-

infectious risk factors [48], this study did not show the association between low relative wealth with functional disability. The lack of association between poverty and disability in this study could be due to the minimal average number of days lost due to disability (5 days/month) which may have had a limited influence on income in this subsistence farming setting.

Our study has several strengths. We investigated a poorly investigated area in relation to the wellbeing of people with epilepsy in LMICs. The use of key informants and health extension workers working at the community level for identification and referral of potential study participants helped to ensure representativeness of the sample and generalizability. We focused on patient-reported outcome measurement, rather than just clinical outcomes, which are important for wellbeing but often overlooked. The diagnosis of comorbid mental disorders was carried out by psychiatric nurses using a gold standard approach. The use of a clinical interview is particularly important in clinical populations where the illness or side effects of medication can lead to erroneously inflated scores on screening tools. There are few studies from sub-Saharan Africa that have used diagnostic verification of mental disorder [49]. One of the main limitations of this study was the cross-sectional design so we cannot infer any temporal association or causality. Even though the epilepsy definition used by mhGAP and ILAE (International League Against Epilepsy) is similar, diagnostic tools like EEG were not used in this study. This was also limited confirmatory diagnosis of focal seizures by the PHC. A further limitation of this study was that antiepileptic drugs and their side effects were not measured which were additional potential confounding factors. It is also possible that co-morbid anxiety disorders were under-diagnosed and/or categorised as depressive disorders by the clinician interviewers.

Overall, co-morbid mental disorders were associated with poorer quality of life and impairment independently of effects on seizure control [12]. The findings of this study have several implications for the clinical evaluation or management of chronic health disorders such as epilepsy. The ultimate treatment goal for PWE, whether they live in a high- or low-income country, is to have total control of the seizure with minimal antiepileptic drug side effects, good quality of life and day-to-day functioning. The achievement of these goals will require a multidisciplinary approach, tackling physical health, mental health, poverty and stigma in an integrated fashion. At the health facility level, indicated screening for comorbid mental disorders in PWE can lead to improved detection of mental disorders. Task-shared models of care are being used to bring mental health care into primary care settings [22]. Thus, healthcare professionals should not manage seizures alone, but must also be attentive to comorbid mental disorders. Furthermore, interventions to address the impact of stigma should be planned and implemented. Future research is recommended to study the trajectory of quality of life over time in relation to comorbid mental disorders and other potential mediators.

## Supporting information

**S1 Dataset.**
(DTA)

## Acknowledgments

We are grateful for the participants and their families, the PRIME project and all its staff.

## Author Contributions

**Conceptualization:** Ruth Tsigebrhan, Abebaw Fekadu, Charles R. Newton, Martin J. Prince, Charlotte Hanlon.

**Data curation:** Ruth Tsigebrhan.

**Formal analysis:** Ruth Tsigebrhan, Girmay Medhin, Martin J. Prince, Charlotte Hanlon.

**Funding acquisition:** Ruth Tsigebrhan, Abebaw Fekadu, Charles R. Newton, Martin J. Prince, Charlotte Hanlon.

**Investigation:** Girmay Medhin.

**Methodology:** Ruth Tsigebrhan, Abebaw Fekadu, Girmay Medhin, Charles R. Newton, Martin J. Prince, Charlotte Hanlon.

**Project administration:** Ruth Tsigebrhan, Abebaw Fekadu.

**Resources:** Ruth Tsigebrhan, Charles R. Newton, Martin J. Prince, Charlotte Hanlon.

**Supervision:** Abebaw Fekadu, Girmay Medhin, Charles R. Newton, Martin J. Prince, Charlotte Hanlon.

**Validation:** Girmay Medhin, Martin J. Prince, Charlotte Hanlon.

**Writing – original draft:** Ruth Tsigebrhan.

**Writing – review & editing:** Ruth Tsigebrhan, Abebaw Fekadu, Girmay Medhin, Charles R. Newton, Martin J. Prince, Charlotte Hanlon.

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
