## [Decision Letter · Decision Letter 0]

18 Jun 2020

PONE-D-20-08958

Comorbid mental disorders and quality of life of people with epilepsy attending primary health care clinics in rural Ethiopia.

PLOS ONE

Dear Dr. Tsigebrhan,

Thank you for submitting your manuscript to PLOS ONE. After careful consideration, we feel that it has merit but does not fully meet PLOS ONE’s publication criteria as it currently stands. Therefore, we invite you to submit a revised version of the manuscript that addresses the points raised during the review process.

We look forward to receiving your revised manuscript.

Kind regards,

Thach Duc Tran, M.Sc., Ph.D.

Academic Editor

PLOS ONE

Journal Requirements:

2. Please include additional information regarding the survey or questionnaire used in the study and ensure that you have provided sufficient details that others could replicate the analyses.

For instance, if you developed a questionnaire as part of this study and it is not under a copyright more restrictive than CC-BY, please include a copy, in both the original language and English, as Supporting Information. Moreover, please include more details on how the questionnaire was pre-tested, and whether it was validated.

Reviewers' comments:

Reviewer's Responses to Questions

**Comments to the Author**

1. Is the manuscript technically sound, and do the data support the conclusions?

Reviewer #1: Yes

Reviewer #2: Yes

2. Has the statistical analysis been performed appropriately and rigorously? 

Reviewer #1: Yes

Reviewer #2: Yes

3. Have the authors made all data underlying the findings in their manuscript fully available?

Reviewer #1: Yes

Reviewer #2: Yes

4. Is the manuscript presented in an intelligible fashion and written in standard English?

Reviewer #1: Yes

Reviewer #2: Yes

5. Review Comments to the Author

Reviewer #1: Very interesting and well conducted study about mental health issues among patients with epilepsy. However, a few minor correction will improve the manuscript and make and even better read.

The manuscript would clearly improve by undergoing anti-verbiage drive. It is full of unnecessary words and repetitions which could be removed without compromising the clarity of the report.

Could the authors please avoids the use of PWE as this is a matter of irritation to many patients with epilepsy, in my experience. Just spell it out in full. Also, the authors should remember that this is report on epilepsy so there is no need to repeat this at each paragraph.

The author don't seem to be cognizant with the concept of keywords. Indeed, any term or word which is already in the title should not be in the list of keywords. Could this be corrected?

Reviewer #2: The authors present a cross-sectional study on the prevalence of mental comorbidities in persons with epilepsy from a region in Ethiopia. The study is of interest for clinicians and researchers especially as it is from an area from which only limited data is available. The two-staged approach of identifying study participants and the methods of diagnostic assessment and data collection are as well of interest. I have only a few comments:

Epilepsy duration should be taken into account.

I understand about half of the population are Muslims bit the vast majority of study participants is of orthodox religion. Could the authors explain? At least they assessed for religion. Do they see any association with the study results?

Table 2 dos not show any data on anxiety disorder: Could the authors explain?

Discussion is a bit lengthy and extends to statements which are certainly common sense (poverty reduction and improved education are necessary) but their association to the study results is rather loose.

The main surprising feature is the smaller proportion of persons with mental comorbidity in comparison to other studies. This might be discussed further. Are there any other factors rather than those discussed by the authors? Do differences in presentation of depression have any consequences for further research?

6. PLOS authors have the option to publish the peer review history of their article (what does this mean?). If published, this will include your full peer review and any attached files.

Reviewer #1: No

Reviewer #2: Yes: Christian Brandt

---

## [Author Response · Author response to Decision Letter 0]

30 Jul 2020

Thank you for the constructive comments and here is the point by point response to the editorial and reviewer comments.

Editor comments

Response 1

This has been now corrected. 

2. Please include additional information regarding the survey or questionnaire used in the study and ensure that you have provided sufficient details that others could replicate the analyses.

Response 2

Additional information regarding the questionnaires used has been included and attached as supporting information (please see manuscript pages 8 and 9). 

Reviewer 1

3. Very interesting and well conducted study about mental health issues among patients with epilepsy. However, a few minor correction will improve the manuscript and make and even better read.

Response 3: Thank you for the encouraging feedback. We hope we have addressed your comments adequately below. 

4. The manuscript would clearly improve by undergoing anti-verbiage drive. It is full of unnecessary words and repetitions which could be removed without compromising the clarity of the report.

Response 4: We have edited the manuscript and attempted to remove unnecessary words and repetitions. 

5. Could the authors please avoids the use of PWE as this is a matter of irritation to many patients with epilepsy, in my experience. Just spell it out in full. Also, the authors should remember that this is report on epilepsy so there is no need to repeat this at each paragraph.

Response 5: We have now corrected this throughout the manuscript. 

6. The author don't seem to be cognizant with the concept of keywords. Indeed, any term or word which is already in the title should not be in the list of keywords. Could this be corrected?

Response 6: We have now corrected the keywords. 

Reviewer 2

7. The authors present a cross-sectional study on the prevalence of mental comorbidities in persons with epilepsy from a region in Ethiopia. The study is of interest for clinicians and researchers especially as it is from an area from which only limited data is available. The two-staged approach of identifying study participants and the methods of diagnostic assessment and data collection are as well of interest. I have only a few comments:

Response 7: Thank you for your helpful comments. 

8. Epilepsy duration should be taken into account.

Response 8

We have rerun the analysis including duration of epilepsy and have included the revised findings in the manuscript. This has not affected the main findings or interpretation. 

9. I understand about half of the population are Muslims but the vast majority of study participants is of orthodox religion. Could the authors explain? At least they assessed for religion. Do they see any association with the study results?

Response 9

In the Gurage zone as a whole, the majority of the population are Muslims, but in the district of Sodo and the data collection sites for this study the majority of the participants are Orthodox Christians. Therefore, our study sample reflected the population from which they were drawn.

We did not include religion in the analysis. Religion is a sensitive topic in this community. Any observed association between religious affiliation and mental ill-health (which is stigmatised) could be misunderstood as being causal, even though such an association could be confounded by other variables (such as socio-economic status). As we had no a priori justification to consider religion as an important explanatory variable for this research question we therefore omitted it from the analysis. 

10. Table 2 does not show any data on anxiety disorder: Could the authors explain?

Response 10

That is correct, there were no participants with anxiety disorders. This was one of the surprising findings of this study. We noted in the discussion that this may reflect the presentation of depression in this setting, which tends to also have anxiety symptoms. The psychiatric nurses may have prioritised the diagnosis of depression over that of anxiety. We have added the following to the discussion:

“The way in which depression and anxiety manifest and the non-biomedical causal attributions of depressive/anxiety symptoms in this socio-cultural context [43] could have contributed to the low detection by mental health professionals applying Western diagnostic criteria [42]. The diagnosis and presentation of depression in Ethiopian culture appears to include a combination of anxiety, somatic and depressive symptoms rather than typical DSM criteria of depression [44]. This might have masked and contributed to the absence of diagnosis of anxiety disorders in this study. Psychiatry nurses could also be biased towards diagnosing more severe conditions like depression than anxiety disorders in their clinical practice.

We have now added this as a potential limitation to the study.

‘It is also possible that co-morbid anxiety disorders were under-diagnosed and/or categorised as depressive disorders by the clinician interviewers.’ 

11. Discussion is a bit lengthy and extends to statements which are certainly common sense (poverty reduction and improved education are necessary) but their association to the study results is rather loose.

The main surprising feature is the smaller proportion of persons with mental comorbidity in comparison to other studies. This might be discussed further. Are there any other factors rather than those discussed by the authors? Do differences in presentation of depression have any consequences for further research?

Response 11

We have now elaborated on the relatively low levels of co-morbid mental illness and noted the relevance for future research.

We have cut some of the sentences which made recommendations that went beyond our study findings.

---

## [Decision Letter · Decision Letter 1]

11 Aug 2020

Comorbid mental disorders and quality of life of people with epilepsy attending primary health care clinics in rural Ethiopia.

PONE-D-20-08958R1

Dear Dr. Tsigebrhan,

We’re pleased to inform you that your manuscript has been judged scientifically suitable for publication and will be formally accepted for publication once it meets all outstanding technical requirements.

Kind regards,

Thach Duc Tran, M.Sc., Ph.D.

Academic Editor

PLOS ONE

Additional Editor Comments (optional):

Reviewers' comments:

Reviewer's Responses to Questions

**Comments to the Author**

1. If the authors have adequately addressed your comments raised in a previous round of review and you feel that this manuscript is now acceptable for publication, you may indicate that here to bypass the “Comments to the Author” section, enter your conflict of interest statement in the “Confidential to Editor” section, and submit your "Accept" recommendation.

Reviewer #1: All comments have been addressed

Reviewer #2: All comments have been addressed

2. Is the manuscript technically sound, and do the data support the conclusions?

Reviewer #1: Yes

Reviewer #2: Yes

3. Has the statistical analysis been performed appropriately and rigorously? 

Reviewer #1: Yes

Reviewer #2: Yes

4. Have the authors made all data underlying the findings in their manuscript fully available?

Reviewer #1: (No Response)

Reviewer #2: Yes

5. Is the manuscript presented in an intelligible fashion and written in standard English?

Reviewer #1: Yes

Reviewer #2: Yes

6. Review Comments to the Author

Reviewer #1: thanks for revising the manuscript; this is now a much better report and i would not have any further comments.

Reviewer #2: (No Response)

7. PLOS authors have the option to publish the peer review history of their article (what does this mean?). If published, this will include your full peer review and any attached files.

Reviewer #1: No

Reviewer #2: **Yes: **Christian Brandt

---

## [Editor Report · Acceptance letter]

30 Sep 2020

PONE-D-20-08958R1 

Comorbid mental disorders and quality of life of people with epilepsy attending primary health care clinics in rural Ethiopia. 

Dear Dr. Tsigebrhan:

I'm pleased to inform you that your manuscript has been deemed suitable for publication in PLOS ONE. Congratulations! Your manuscript is now with our production department. 

Kind regards, 

on behalf of

Dr. Thach Duc Tran 

Academic Editor

PLOS ONE